# Effects of Rice Husk Biochar and Compost Amendments on Soil Phosphorus Fractions, Enzyme Activities and Rice Yields in Salt-Affected Acid Soils in the Mekong Delta, Viet Nam

**Doan Thi Truc Linh** [1,2], **Chau Minh Khoi** [2], **Karl Ritz** [3], **Nguyen Van Sinh** [2], **Nguyen Thi Kim Phuong** [2], **Huynh Mach Tra My** [1,2], **Tran Ba Linh** [2], **Dang Duy Minh** [2], **Thi Tu Linh** [4] and **Koki Toyota** [1,*]

1 Graduate School of Bio-Applications and Systems Engineering, Tokyo University of Agriculture and Technology, Tokyo 184-8588, Japan; truclinh@ctu.edu.vn (D.T.T.L.); hmtmy@ctu.edu.vn (H.M.T.M.)
2 College of Soil Science, Can Tho University, Campus II, Can Tho 900100, Vietnam; cmkhoi@ctu.edu.vn (C.M.K.); nvsinh@ctu.edu.vn (N.V.S.); ntkimphuong@ctu.edu.vn (N.T.K.P.); tblinh@ctu.edu.vn (T.B.L.); ddminh@ctu.edu.vn (D.D.M.)
3 School of Biosciences, University of Nottingham, Sutton Bonington LE12 5RD, UK; karl.ritz@nottingham.ac.uk
4 Agricultural Extension Center of Kien Giang Province, Rach Gia 9200000, Vietnam
* Correspondence: kokit@cc.tuat.ac.jp; Tel.: +81-42-388-7915

**Abstract:** Given that rice husk biochar has been shown to modulate salinity in salt-affected acid soils, the objective of this study was to investigate the effects of organic amendment of salinized acid soils on P fractions, enzyme activities, and associated rice yield. Four treatments, viz. Rice–Rice–Rice, [RRR]; Fallow–Rice–Rice, [FRR]; Fallow–Rice–Rice + 3 Mg ha$^{-1}$ of compost [FRR + Comp]; and Fallow–Rice–Rice + 10 Mg ha$^{-1}$ of biochar [FRR + BC] were established at Ben Tre and Kien Giang sites, Viet Nam, over six consecutive crops. Soil properties at harvest of the sixth crop showed that there were diverse patterns of fractionation between P forms with respect to treatment. Overarchingly, biochar increased labile and moderately labile inorganic P and organic P by 30% to 70%, respectively, whilst compost had a relatively modest effect on these pools. Soil phosphatase activities at crop tillering increased following the FRR + Comp and FRR + BC treatments compared with those in RRR, except for acid phosphatase at Ben Tre. At harvest, there were no significant differences between the enzyme activities among the treatments. Rice yield was positively correlated with the more labile forms of P, soil C, and acid phosphatase activity. In the absence of organic amendments, there was no effect of triple versus double rice crops being grown in one-year cycle. Repeated application of biochar (10 Mg ha$^{-1}$ × 5 times) showed potential to increase grain yields and total soil C in salt-affected acid soils, via modulation of P transformations to more plant-available forms.

**Keywords:** organic amendments; phosphatase activity; more labile forms of P; rice yield



## 1. Introduction

In Viet Nam, acid sulfate soil (ASS) covers one million ha [1] in which paddy rice (*Oryza sativa*) is the main crop. One of the characteristics of the conventional farming in ASS is the application of large quantities of inorganic P (Pi) fertilizers to soils to increase crop yields. However, such Pi tends to be rendered unavailable in soils due to chemically based binding phenomena, especially in acidic soils like ASS, since P is strongly adsorbed to Al and Fe hydroxides when the pH is low [2]. Phuong et al. [3] reported that only 12% of total P forms were labile in ASS in the Vietnamese Mekong delta (MD). Amending acid soil with P-containing organic materials such as biochar or compost can increase total P and improve its availability to plants. The associated mechanisms are postulated to be related to the increased relative proportions of differentially available P forms in these materials and interactions with associated soil [4,5]. The associated hierarchy of putative P availability to plants and soil biota can be characterized by the Hedley method [6,7]. In the procedure,

different fractions of P that are bound with complex inorganic and organic compounds in soil are sequentially extracted using a series of solutions with different composition and strength targeting displacement of increasingly bound P forms. The technique has been shown to reveal contrasting patterns of P forms arising from particular fertilization regimes. For instance, analysis by in vitro incubation studies highlighted that application of rice husk biochar to ASS increased $NaHCO_3$ extractable Pi (labile P), while NaOH-Pi (moderate labile P) was decreased because Fe and Al-bound P were formed less under increased pH conditions caused by biochar application [3]. Other studies showed that the application of chemical fertilizer and pig manure to soil resulted in the build-up of organic P fraction (Po) and increased available P [8]. Noticeably, rice yield has been increased by organic fertilizer application, through enhancing available P [9–11]. Among organic amendments, biochar and compost are two common materials [12]. Thus, it is important to explore how biochar or compost addition affects the forms and availability of P remaining in the soil after application.

Soil labile C and available N are considered to be sensitive indicators of changes in the soil C and N pools in response to management practices. Soil labile C is suitable to detect management practices that promote soil C sequestration, making it a useful indicator of soil quality [13,14]. Productivity and functionality in the soil ecosystem are affected by soil labile C, through effects on soil microbes [15]. Therefore, changes in C fractions (particularly the labile fraction) and available N may reflect those in soil fertility and productivity more promptly than other soil properties [16].

Soil enzymes such as β-glucosidase (GLU), acid phosphatase (ACP), alkaline phosphatase (ALP), and urease activities are known to play important roles in C, P, and N cycling, respectively [17]. For example, biochar increased GLU, which may be associated with the increase in C mineralization [18]. Increases in soil phosphatase activities were also observed under organic amendments. For instance, biochar has been shown to increase phosphatase activity in both soil incubation studies [19] and field experiments [20]. By contrast, other research has reported no effect [21] or a reduction in phosphatase activity in biochar-treated field plots [22], suggesting that the combination of biochar and soil properties plays a crucial role in regulating changes in P processes. Huang et al. [23] reported increased urease activity with increasing biochar rates of wheat straw biochar, while the addition of biochar resulted in a significant decrease in soil urease activity [24]. These contrasting results suggest that biochar-induced changes in soil enzyme activities depend on various parameters, such as the feedstock type used, the pyrolysis condition, and biochar application rate [25,26]. Therefore, it is desirable to improve our understanding of the potential effects of biochar and compost with varying properties on the soil enzyme activities in order to enhance the efficacy of biochar and compost applications in agricultural systems.

Our previous study focusing on the Vietnamese MD [27] found that rice husk biochar improved the availability of potassium and reduced the concentration of sodium in salt-affected soil. However, there has been limited long-term experimental research that has investigated the effects of fertilization on rice productivity, as well as P fractions and enzyme activities in salt-affected fields in the MD. Phosphorus often limits plant productivity, especially in tropical regions [28,29]. Whilst application of biochar or compost to soils increases readily extractable P, little is known about how biochar or compost addition affects the other forms of P in the soil after their continuous application in the MD. Since the P fractionation patterns in soil certainly change with respect to types of organic amendment [3,30–32], it was hypothesized that the addition of biochar or compost would alter some biochemical properties and consequently result in an increase in rice yield in the salt-affected acid soils. Therefore, the objectives of this study were (1) to evaluate the effects of rice husk biochar and compost amendments on the amounts of various P pools, labile C, available N, and enzyme activities in salt-affected acid soils and (2) to determine the factors affecting rice grain yield.

## 2. Materials and Methods

### 2.1. Materials Description

Phuong et al. [26] have already reported the key properties of the soils and organic amendments utilized in this study. In summary, the commercially available compost (Phan Huu Co Sinh Hoc Nha Nong PPE, PPE Co., Ltd., Can Tho, Vietnam) was made from sugarcane filter cake and had a high concentration of soluble Ca and exchangeable Ca, while the commercially available biochar (Mai Anh Co., Dong Thap, Vietnam, the iodine number 152 mg g$^{-1}$) was made from rice husk at 700 °C and had a low concentration of these properties (Table 1). Soil texture at both sites is silty clay with broadly similar basic properties (pH, EC, total C, total N, soluble Ca, exchangeable Ca), while available P content was contrasting (Table 1).

**Table 1.** Chemical properties of soil and amendments used in the experiment.

| | Organic Materials | | Soils | |
|---|---|---|---|---|
| | **Compost** | **Biochar** | **Ben Tre** | **Kien Giang** |
| pH (H$_2$O) * (1:2.5) | | | 4.64 | 4.65 |
| pH (H$_2$O) * (1:5) | 8.7 | 7.7 | | |
| EC * (mS cm$^{-1}$) (1:2.5) | | | 1.24 | 1.16 |
| EC * (mS cm$^{-1}$) (1:5) | 17.1 | 4.1 | | |
| Total C * (g kg$^{-1}$) | 154 | 471 | 15.7 | 15.0 |
| Labile C (g kg$^{-1}$) | 3.13 | 1.18 | 0.59 | 0.39 |
| Total N * (g kg$^{-1}$) | 26 | 4.72 | 1.55 | 1.32 |
| Available N (mg kg$^{-1}$) | 528 | 8.13 | 24.7 | 22.5 |
| Available P * (mg kg$^{-1}$) | 3600 | 800 | 6.91 | 20.9 |
| Soluble K * (mg kg$^{-1}$) | 20 | 3.35 | 0.15 | 0.07 |
| Exchangeable K * (mg kg$^{-1}$) | 15 | 12.9 | 0.63 | 0.52 |
| Soluble Ca * (mg kg$^{-1}$) | 7.29 | 0.15 | 1.70 | 0.96 |
| Exchangeable Ca * (mg kg$^{-1}$) | 61.6 | 0.16 | 6.01 | 3.44 |

* Cited from Phuong et al. [27]. pH values were measured with an electrode. Total C and N were measured with the combustion method. Available P was measured with the Bray II method. Soluble K and Ca were extracted with deionized water. Exchangeable K and Ca were extracted with 0.1 M BaCl$_2$ solution and then soluble K and Ca were subtracted. Both ions were measured with a flame photometer. Labile C and available N were measured with the methods described in Section 2.3.1.

Initial values for the P fractions in soils prior to the commencement of amendment regimes are given in Table 2. Those for biochar and compost are given in Phuong et al. [3]. In summary, the total P content was 10 times lower in biochar (1.1 g kg$^{-1}$) than in compost (11.5 g kg$^{-1}$), the percentage of available Pi (H$_2$O-Pi and NaHCO$_3$-Pi) contents of total P were greater in biochar (25%) compared to compost (13%). By contrast, the NaOH-Pi and HCl-Pi contents were 4.0 and 2.3 times greater, respectively, in compost than in biochar.

**Table 2.** Phosphorus contents (mg kg$^{-1}$) in different P fractions in the study soils at the onset of experiment.

| P Fractions | H$_2$O-Pi | NaHCO$_3$-Pi | NaHCO$_3$-Po | NaOH-Pi | NaOH-Po | H$_2$SO$_4$-Pi | Residual P |
|---|---|---|---|---|---|---|---|
| Ben Tre | 0.48 | 26.3 | 65.5 | 78.1 | 62.9 | 35.6 | 219 |
| Kien Giang | 0.73 | 36.0 | 70.5 | 146 | 72.5 | 54.2 | 216 |

Pi: inorganic form of phosphorus, Po: organic form of phosphorus.

### 2.2. Experimental Designs

The field trial was located in Thanh Phu district, Ben Tre province (9°58′22.51″ N, 106°28′51.22″ E) and U Minh Thuong district, Kien Giang province (9°43′34.43″ N, 105°10′55.06″ E) in the Vietnamese Mekong delta [26]. According to Hydrometeorological Station of Ben Tre and Kien Giang province, the rainfall from July 2019 to January 2020 tended

to decrease, with no rain in January at either site (Tables S1 and S2). The four treatments were applied over two years in a randomized complete block design with four replicate plots, each 8 m × 5 m (length × breadth). Each plot was separated by bunds (30 cm wide × 40 cm high) in order to minimize seepage between adjacent plots and from the surrounding field. Treatments comprised: (i) three times rice cultivation in a year (RRR), (ii) two times rice cultivation with fallow in spring–summer crop (FRR), (iii) FRR plus compost at 3 Mg ha$^{-1}$ crop$^{-1}$ (FRR + Comp), and (iv) FRR plus biochar at 10 Mg ha$^{-1}$ crop$^{-1}$ (FRR + BC). The experiment spanned two years and involved six consecutive cropping cycles. This study focuses on the Winter Spring 2019–2020 (WS 19–20) phase of the experiment, as the details of the experiment up to the Summer Autumn 2019 phase have already been provided in Phuong et al. [26]. For instance, *Oryza sativa* L. c.v. OM 5451 was planted at a rate of 150 kg ha$^{-1}$ and direct sowing was conducted on 4th October for Kien Giang. Transplanting of local variety was done on 15th September at Ben Tre. As a consequence of severe water shortage, the rice in the 1st crop only grew for a period of 1.5 months and died. Therefore, biochar and compost were not applied in the subsequent crop. Consequently, the total accumulative mass of biochar and compost amendments applied over the two years were 50 Mg ha$^{-1}$ (10 Mg ha$^{-1}$ × 5 times) and 15 Mg ha$^{-1}$ (3 Mg ha$^{-1}$ × 5 times), respectively.

Biochar was first spread at a rate of 10 Mg ha$^{-1}$ and then mixed thoroughly in the topsoil (0 to 15 cm) using a plowing machine four days prior to transplanting or sowing. The compost was also spread over the soil surface at a rate of 3 Mg ha$^{-1}$ as a basal dressing, as the same day with transplanting or sowing. Flooded conditions were maintained during the rice growing period, and irrigation was stopped around 15 days before harvest. Manual weeding and herbicide application were performed throughout the experiment. Insecticide regimes according to the local practice were used during the period of crop growth. Rice straw was removed after harvest for each crop.

The inorganic NPK fertilizer was applied with total amounts of 100 kg N ha$^{-1}$, 60 kg P$_2$O$_5$ ha$^{-1}$, and 30 kg K$_2$O ha$^{-1}$. Nitrogen was applied as urea, split into 20%, 40%, and 40% of the total N fertilizer at 7, 20, and 40 days after transplanting or sowing, respectively. Superphosphate (16% P$_2$O$_5$) was applied once before transplanting or sowing. Potassium was applied as two equal doses of KCl (60% K$_2$O) at 20 and 40 days after transplanting or sowing.

In WS 19–20 (6th crop in RRR and 4th crop in FRR, FRR + Comp, FRR + BC), soil samples were collected from a depth of 0 to 20 cm at the stage of the tillering on 4th and 18th November and harvesting on 24th December 2019 and 4th January 2020 of Kien Giang and Ben Tre, respectively. Soil from five different random points was mixed in each replicate plot. In order to assess the chemical and biological properties of the soil, the soil samples were dried naturally at room temperature and then passed through a 0.5 mm sieve. Soil samples at the tillering stage (40 days) were used for the determination of enzyme activities (GLU, ACP, ALP, and urease). At the harvest, the soil samples were analyzed for pH, total C, labile C, total N, available N, different P fractions, as well as enzyme activities.

At harvest in WS 19–20, two replicates of sub-plot (0.25 m$^2$: length × width, 0.5 m × 0.5 m) were randomly set up in each plot and rice plants were cut at the ground level and separated into straw and grain. Then, the grains of each plot were dried in an oven at 105 °C and weighed to determine the dry biomass production of the grain.

## 2.3. Sample Analysis

### 2.3.1. Soil Chemical Properties

Soil P was fractionated via sequential chemical extraction following the fractionation scheme based on the Hedley method [6] as modified by Zhang et al. [33]. In brief, various extractants were used to separate the soil P into different pools including labile P (H$_2$O-Pi, NaHCO$_3$-Pi, NaHCO$_3$-Po), moderate labile (NaOH-Pi, NaOH-Po, H$_2$SO$_4$-Pi), and non-labile P (Residual P). First, 0.5 g of air-dried soil was extracted with 30 mL of deionized water and then sequentially extracted with 30 mL of 0.5 M NaHCO$_3$ at a pH of 8.5, 0.1 M

NaOH, and 30 mL of 0.5 N $H_2SO_4$. At each extraction step, suspensions consisting of soil and extractant were shaken at 120 rpm for 16 h, centrifuged for 10 min at $8000 \times g$, and then filtered through Advantec No.5C filter paper. The filtrates were used for the determination of Pi. The total P of 0.5 M $NaHCO_3$ and 0.1 M NaOH extracts were determined after treating these extracts with 4 mL condensed $H_2SO_4$ and 1 mL $HClO_4$ at 250 °C until the extracts became transparent. The Po of 0.5 M $NaHCO_3$, 0.1 M NaOH was calculated by the difference between total P and Pi in the same filtrates. The final step included complete digestion with $H_2SO_4$–$HClO_4$ (1:2, v:v) to obtain residual P. P concentrations in all filtrates and digests were determined by the molybdenum blue colorimetric method [34].

Soil pH ($H_2O$) was measured by mixing soil with deionized water at a ratio of 1:2.5 (soil:water, w:v) for 1 h at 120 rpm and by using a pH meter by Hanna–HI 8314. Total C and total N content were analyzed by CN corder apparatus (MT-700, Yanaco Co., Tokyo, Japan). Labile C content was determined by the method of Weil et al. [35]. Non-labile C was calculated by the difference between total C and labile C. For available N, 2.0 g soil was extracted with 10 mL of phosphate buffer, and the supernatant was measured for the absorbance of 260 nm [36].

### 2.3.2. Soil Enzyme Activities

Several studies have reported that there were no significant differences in GLU, ACP, ALP, and urease activities between field-moist and air-dried samples [37–41]. Therefore, enzyme activities were measured using air-dried soil samples. The methods of GLU, ACP, and ALP activities were followed by Tabatabai [42]. The measurement of urease activity was carried out following the method by Nannipieri et al. [43].

### 2.4. Statistical Analysis

The effects of biochar or compost amendments on the considered parameters were assessed using one-way and two-way ANOVA via SPSS version 20.0, using a significance threshold of 5%. A principal component analysis (PCA) was performed using Statistic 7 for the chemical properties and enzyme activities of all treatments.

### 3. Results

### 3.1. Effects of Biochar and Compost Amendments on Chemical Properties of Fertilized Salt-Affected Acid Soils

### 3.1.1. pH, Total C, Labile C, Non-Labile C, Total N, and Available N

The pH levels of the study soils indicated acidity, ranging from 5.0 to 5.2 for Ben Tre and from 4.9 to 5.0 for Kien Giang (Table 3). Nevertheless, no notable variations in soil pH were observed among the treatments. Total C and non-labile C content was significantly greater in FRR + BC than in other treatments at two sites. However, no significant difference was detected in soil labile C among the treatments. Similarly, total N and available N were not affected by compost and biochar amendments. No significant interaction was found between site and treatment in terms of the level of total C, labile C, non-labile C, total N, and available N.

**Table 3.** Effect of compost and biochar amendments on total C, labile C, non-labile C, total N, and available N in fertilized salt-affected acid soils at Ben Tre and Kien Giang sites.

| Sites (S) | Treatments (T) | pH ($H_2O$) | Total C | Labile C | Non-Labile C | Total N | Available N |
|---|---|---|---|---|---|---|---|
| | | | g C kg$^{-1}$ | g C kg$^{-1}$ | g C kg$^{-1}$ | g N kg$^{-1}$ | mg N kg$^{-1}$ |
| Ben Tre | RRR | 5.05 | 16.3 b | 0.55 | 15.8 b | 2.03 | 22.3 |
| | FRR | 5.21 | 16.2 b | 0.54 | 15.7 b | 1.98 | 19.0 |
| | FRR + Comp | 4.97 | 17.9 b | 0.54 | 17.3 b | 2.09 | 23.8 |
| | FRR + BC | 5.18 | 23.3 a | 0.56 | 22.7 a | 2.23 | 20.8 |

**Table 3.** *Cont.*

| Sites (S) | Treatments (T) | pH ($H_2O$) | Total C | Labile C | Non-Labile C | Total N | Available N |
|---|---|---|---|---|---|---|---|
| | | | g C kg$^{-1}$ | g C kg$^{-1}$ | g C kg$^{-1}$ | g N kg$^{-1}$ | mg N kg$^{-1}$ |
| Kien Giang | RRR | 4.85 | 19.7 b | 0.55 | 19.1 b | 2.15 | 19.5 |
| | FRR | 4.91 | 19.7 b | 0.52 | 19.2 b | 2.13 | 20.9 |
| | FRR + Comp | 5.04 | 17.4 b | 0.49 | 16.9 b | 1.97 | 20.5 |
| | FRR + BC | 4.96 | 25.7 a | 0.53 | 25.2 a | 2.11 | 21.9 |
| ANOVA | | | | | | | |
| S | | ns | * | ns | * | ns | ns |
| T | | ns | *** | ns | *** | ns | ns |
| S × T | | ns | ns | ns | ns | ns | ns |

Data (mean, *n* = 4) followed by different letters indicate a significant difference among treatments at two sites by Duncan test at *p* < 0.05; pooled standard error (SE = 0.76); RRR, Rice–Rice–Rice; FRR, Fallow–Rice–Rice; FRR + Comp, Fallow–Rice–Rice + 3 Mg ha$^{-1}$ of compost; FRR + BC, Fallow–Rice–Rice + 10 Mg ha$^{-1}$ of biochar, * *p* < 0.05; *** *p* < 0.001; ns, not significant.

### 3.1.2. P Fractionation

Biochar amendments generally increased forms of Pi at both sites, excepted $H_2O$-Pi and $H_2SO_4$-Pi at Ben Tre site (Table 4). Interaction between site and treatment was only significant in NaHCO$_3$-Po and NaOH-Po (*p* < 0.05). All P fractions differed significantly between sites, except for NaOH-Po. However, treatment effects were only significant in relation to NaHCO$_3$-Pi, NaOH-Pi, and $H_2SO_4$-Pi.

**Table 4.** Effect of compost and biochar amendments on labile P pool, moderately labile P pool, and non-labile P pool in fertilized salt-affected acid soils at Ben Tre and Kien Giang sites.

| Sites | Treatments | Labile P | | | Moderately Labile P | | | Non-Labile P |
|---|---|---|---|---|---|---|---|---|
| | | $H_2O$-Pi | NaHCO$_3$-Pi | NaHCO$_3$-Po | NaOH-Pi | NaOH-Po | $H_2SO_4$-Pi | Residual-P |
| | | | | | mg kg$^{-1}$ | | | |
| Ben Tre | RRR | 0.38 | 14.1 b | 30.2 | 83.7 b | 32.6 b | 38.2 | 232 |
| | FRR | 0.41 | 17.3 b | 34.4 | 88.8 b | 37.0 ab | 44.3 | 220 |
| | FRR + Comp | 0.33 | 31.0 ab | 31.5 | 126 ab | 54.9 a | 55.5 | 245 |
| | FRR + BC | 0.41 | 48.8 a | 24.3 | 133 a | 50.1 ab | 51.1 | 232 |
| Kien Giang | RRR | 0.90 ab | 30.7 b | 82.5 b | 133 b | 47.8 b | 55.4 b | 209 ab |
| | FRR | 0.86 b | 33.6 b | 82.7 b | 131 b | 74.4 a | 55.3 b | 193 b |
| | FRR + Comp | 1.18 ab | 42.3 b | 71.4 b | 143 b | 44.3 b | 60.5 b | 229 a |
| | FRR + BC | 1.39 a | 66.2 a | 123 a | 192 a | 39.5 b | 78.1 a | 209 ab |
| ANOVA | | | | | | | | |
| S | | *** | * | *** | *** | ns | *** | * |
| T | | ns | *** | ns | *** | ns | ** | ns |
| S × T | | ns | ns | * | ns | * | ns | ns |

Data (mean, n = 4) followed by different letters indicate a significant difference among treatments at two sites by Duncan test at *p* < 0.05; SE, pooled standard error (SE = 4.90); RRR, Rice–Rice–Rice; FRR, Fallow–Rice–Rice; FRR + Comp, Fallow–Rice–Rice + 3 Mg ha$^{-1}$ of compost; FRR + BC, Fallow–Rice–Rice + 10 Mg ha$^{-1}$ of biochar. * *p* < 0.05; ** *p* < 0.01; *** *p* < 0.001; ns, not significant.

### 3.1.3. Effect on Labile P Pools

$H_2O$-Pi concentrations were significantly greater in FRR + BC than in FRR at Kien Giang, and those for NaHCO$_3$-Pi were significantly greater in FRR + BC than in RRR and FRR at both sites (Table 4). In comparison with RRR, the NaHCO$_3$-Pi in FRR + BC was increased by 247% and 116% at Ben Tre and Kien Giang, respectively. By contrast, the NaHCO$_3$-Pi in FRR + Comp was found to increase by 121% and 25% at Ben Tre and Kien Giang, respectively. The NaHCO$_3$-Po was significantly greater in FRR + BC at Kien Giang.

By contrast, there were no significant differences in NaHCO$_3$-Po among the treatments at Ben Tre.

### 3.1.4. Effect on Moderately Labile P Pools (NaOH-Pi, NaOH-Po, and H$_2$SO$_4$-Pi)

The concentrations of NaOH-Pi in all treatments were much greater than those of NaOH-Po and H$_2$SO$_4$-Pi (Table 4). The NaOH-Pi was significantly greater in FRR + BC than in FRR and RRR of both sites. There was no significant difference between NaOH-Pi of FRR + Comp and FRR at both sites. Notably, the increase in NaOH-Pi of compost was significantly less than that of biochar at Kien Giang. The NaOH-Po significantly increased more in FRR + Comp than in RRR at Ben Tre. However, there was a significant decrease in FRR + BC and FRR + Comp compared to FRR at Kien Giang. In addition, concentrations of H$_2$SO$_4$-Pi were significantly greater in FRR + BC than in RRR and FRR at Kien Giang, while no significant changes were observed among the treatments at Ben Tre.

### 3.1.5. Effect on Non-Labile P Pools (Residual P)

The biochar and compost amendments did not affect the residual P in soil at Ben Tre. By contrast, residual P content was significantly greater in FRR + Comp than in FRR at Kien Giang (Table 4).

### 3.2. Effects of Biochar and Compost Amendment on Enzyme Activities of Fertilized Salt-Affected Acid Soils

At the tillering stage of rice, there were no significant differences in GLU and urease activities at both sites among the treatments (Figure 1). In comparison to RRR, a significant increase of ACP was observed in FFR + BC and FRR + Comp at Kien Giang by 33% and 34%, respectively. Notably, the significantly greatest ALP by 60% was observed in FRR + Comp at Ben Tre compared with that in RRR. The ALP in FRR + BC was only significantly greater than that in RRR by 56% and 53% at Ben Tre and Kien Giang, respectively. The ALP across all the treatments was two times lower in ACP at both sites. At harvesting stage, the effect of biochar and compost on GLU, ACP, ALP, and urease activities were not significant at either site.

### 3.3. Principal Component Analysis

In the principal component analysis incorporating the suite of soil chemical and enzyme activities, PC1 and PC2 jointly accounted for 46% of the variation (Figure 2). There were no significant site and treatment interactions in relation to these PCs but highly significant main site and treatment effects for both PCs ($p < 0.001$). FRR + BC ordinated distinctly from all other treatments with respect to PC1 and to RRR with respect to PC2, with all other treatments clustering (Figure 2a). The loadings indicated that this separation was driven predominantly by chemical rather than enzyme parameters, notably a combination of soil P fractions (Figure 2b). The site-based separation was driven by the same parameters (Figure 2a inset).

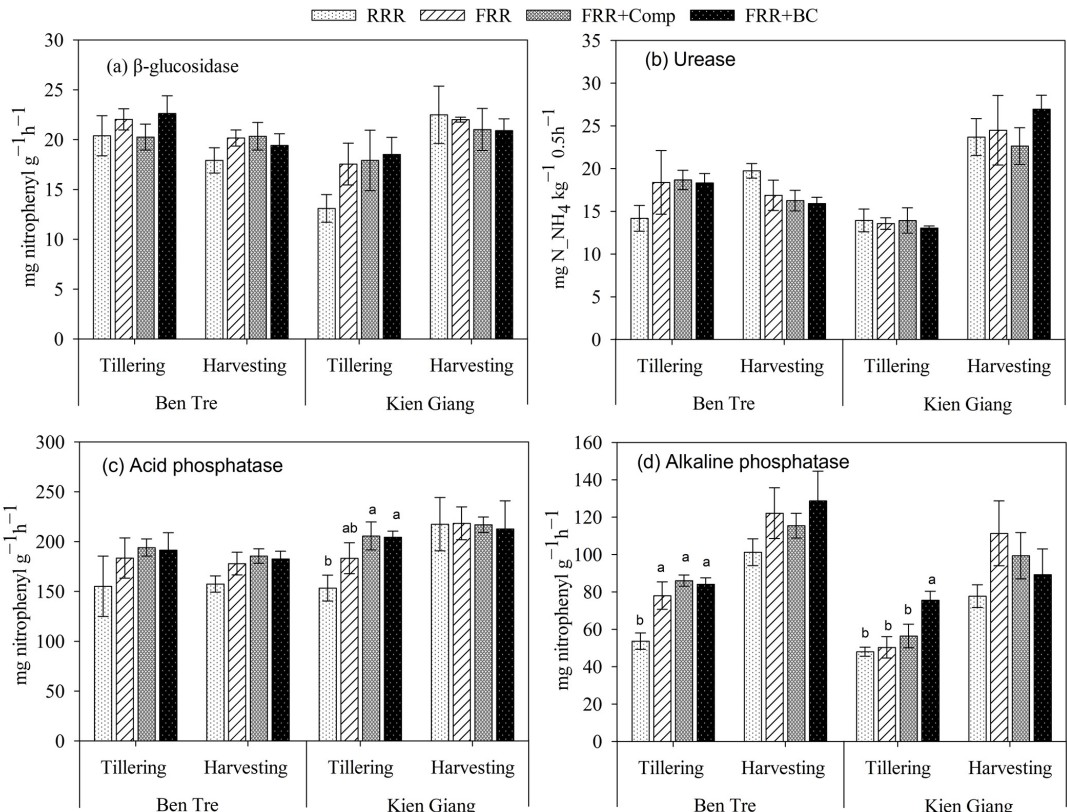

**Figure 1.** Effect of compost and biochar amendments on (**a**) β-glucosidase, (**b**) urease, (**c**) acid phosphatase, and (**d**) alkaline phosphatase activities in salt-affected acid soils at Ben Tre and Kien Giang sites. Error bars indicate the standard error of four replications ($n = 4$). Different letters indicate a significant difference among treatments by Duncan test at $p < 0.05$; ns, not significant; RRR, Rice–Rice–Rice; FRR, Fallow–Rice–Rice; FRR + Comp, Fallow–Rice–Rice + 3 Mg ha$^{-1}$ of compost; FRR + BC, Fallow–Rice–Rice + 10 Mg ha$^{-1}$ of biochar.

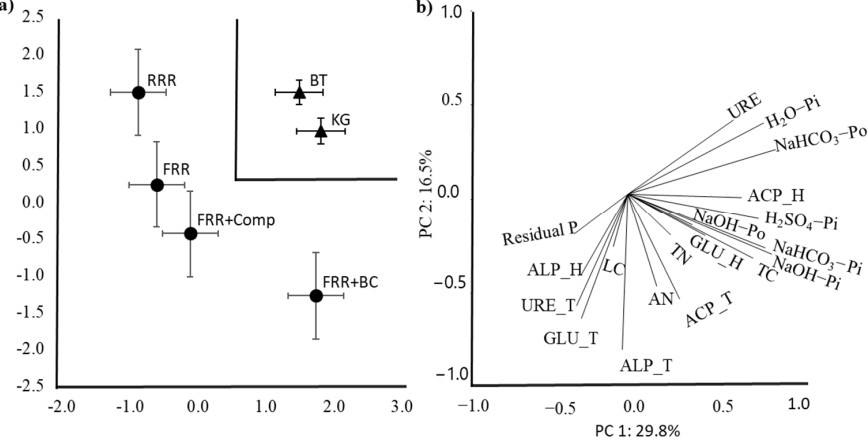

**Figure 2.** Principal component analysis (PCA) based on the soil chemical and biological properties in relation to (**a**) treatments averaged across Ben Tre (BT) and Kien Giang (KG) site treatment, inset shows site ordinations averaged across treatments, (**b**) loadings of each parameter. GLU, β-glucosidase; ACP, Acid phosphatase; ALP, Alkaline phosphatase; URE, urease; Pi, inorganic phosphorus; Po, organic phosphorus; Residual P, Residual phosphorus; LC, labile carbon; AN, available nitrogen. RRR, Rice–Rice–Rice; FRR, Fallow–Rice–Rice; FRR + Comp, Fallow–Rice–Rice + 3 Mg ha$^{-1}$ of compost; FRR + BC, Fallow–Rice–Rice + 10 Mg ha$^{-1}$ of biochar. Data of chemical properties at the harvest, and of biological properties at the tillering (*_T) and harvesting (*_H).

### 3.4. Relationships to Grain Yield

There were no significant site and treatments interaction with respect to grain yield. Biochar amendment significantly increased the grain yield of the rice crop compared with no amendment at both sites (Figure 3). The FRR + BC significantly increased by 0.68 Mg ha$^{-1}$ and 0.76 Mg ha$^{-1}$ compared with FRR at Ben Tre and Kien Giang, respectively. By contrast, FRR + Comp only significantly increased at Ben Tre compared to FRR.

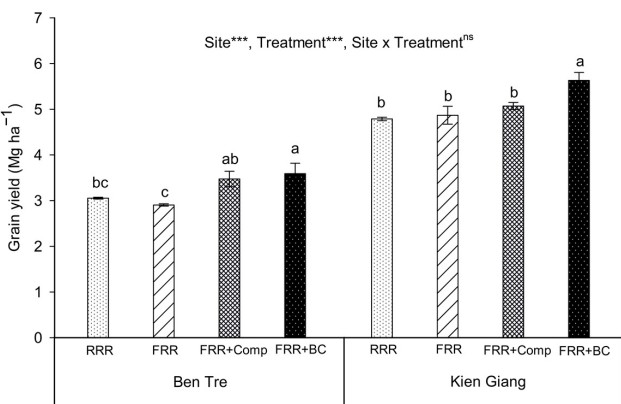

**Figure 3.** Effect of compost or biochar amendment on rice grain yield on the 4th (FRR) and 6th (RRR) harvest at Ben Tre and Kien Giang sites. Error bars indicate the standard error of four replications (n = 4). Different letters indicate a significant difference among treatments by Duncan test at $p < 0.05$; ns, not significant; *** $p < 0.001$; RRR, Rice–Rice–Rice; FRR, Fallow–Rice–Rice; FRR + Comp, Fallow–Rice–Rice + 3 Mg ha$^{-1}$ of compost; FRR + BC, Fallow–Rice–Rice + 10 Mg ha$^{-1}$ of biochar.

Regression analysis showed that there were significant linear relationships between grain yield and the more labile P fractions, as well as total C and ACP activity (Figure 4). The greatest correlation was observed for grain yield and labile P.

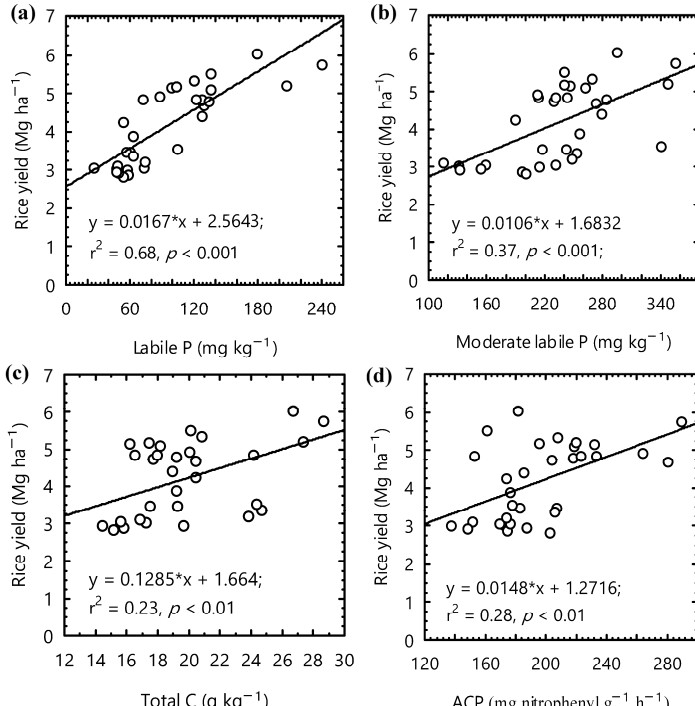

**Figure 4.** Linear regression between (**a**) labile and (**b**) moderately labile P fractions; (**c**) total soil C; (**d**) acid phosphatase (ACP) and rice grain yield. P values relate to significance of regression coefficient. * in the equation indicates a significant ($p < 0.05$) slope value.

### 3.5. Triple versus Double Rice Cropping

In the absence of the organic amendments, there were no significant differences manifest between RRR versus FRR with respect to any of the measured parameters (Tables 3 and 4; Figures 1–3), with the single exception of alkaline phosphatase at Ben Tre at the rice tillering stage (Figure 1c).

## 4. Discussion

*4.1. Response of P Fractions, Labile C, and Available N to Biochar or Compost Amendments to Fertilized Salt-Affected Acid Soils*

The lowest concentrations of labile P ($H_2O$-Pi and $NaHCO_3$-Pi) measured in all treatments will likely have arisen as a result of crop uptake of P and subsequent removal through rice harvest, since P losses via run off are inconsequential in this field system [44]. The $H_2O$-Pi and $NaHCO_3$-Pi concentrations tended to be the greatest in FRR + BC and followed by FRR + Comp, FRR, and RRR at both sites. These data are consistent with previous studies, which found the significant increase in the labile P pool which caused by application of the biochars derived from sewage sludge and chicken manure feedstock [32], rice husk [3], or animal and sludge-based materials [45]. Another study showed that a sewage sludge biochar and a chicken manure biochar increased $H_2O$-Pi by 77% and 206% and $NaHCO_3$-Pi by 200% and 188%, respectively [32]. Biochar or compost amendments can increase P in the soil through several mechanisms. Firstly, the labile P as the sum of $H_2O$-Pi and $NaHCO_3$-P was incorporated through inputs of biochar (10 Mg ha$^{-1}$) or compost (3 Mg ha$^{-1}$) and those were estimated to be 4.0 kg and 3.3 kg labile P, respectively, in this study. Therefore, the direct input of P from biochar or compost could increase the content of labile P in soils. Secondly, biochar amendment to acid soils decreased P sorption and thus increased P release [46–48]. Thirdly, the addition of biochar may have hastened the conversion of Po to Pi and thereby provided labile P [19,49]. Indeed, an increase in $NaHCO_3$-Pi and a decrease in $NaHCO_3$-Po were observed at Ben Tre (Table 3). Additionally, our findings showed that the application of biochar stimulated ACP activity in soil at tillering, suggesting the enhanced conversion of Po into Pi. Furthermore, a part of $H_2SO_4$-Pi in biochar or compost was transformed into more available P at the low pH values of acidic soil (Table 3), like previous studies [49,50]. These mechanisms may account for why biochar and compost amendment increased soil P availability.

In Kien Giang, the moderately labile P pool was the largest followed by non-labile P and labile P. Like labile P, the NaOH-Pi pool was increased by biochar and compost amendments. This result was consistent with a previous study which reported that the NaOH-Pi pool in Haplic Luvisol increased by wheat straw biochar after 30 days of incubation [49]. However, this finding was contrary to other studies [3,49], which reported that application of biochar inhibited P to precipitate with Fe and Al under increased pH conditions, and this led to decrease NaOH-Pi in the soil. This phenomenon was not the case in our study, as the soil pH was not significantly affected by biochar (Table 3). Besides, Bernerd et al. [51] showed that the soils are naturally acidic with a pH value less than 6.0 and P availability is primarily controlled by Al and Fe. Consequently, after rice uptake, the content of NaOH-Pi was greater than that of $H_2SO_4$-Pi, although the contents of NaOH-Pi in both of biochar and compost were lower than those of $H_2SO_4$-Pi. Biochar and compost amendments at 10 Mg ha$^{-1}$ and 3 Mg ha$^{-1}$ corresponded to the addition of 1.38 kg ha$^{-1}$ and 9.95 kg ha$^{-1}$ of a sum of NaOH-Pi and NaOH-Po into the soil, respectively, which directly increased the soil P content and affected the NaOH-Pi and NaOH-Po fractions.

The $H_2SO_4$-Pi is a phosphate fraction fixed with $Ca^{2+}$ [52]. Biochar and compost were a source of Ca-Pi and compost contained five times greater Ca-Pi than biochar [3]. A significant increase of $H_2SO_4$-Pi was only observed in FRR + BC at Kien Giang. This was consistent with Hong and Lu [52] who found that biochar application significantly increased Ca-Pi concentration in acid soil. Phuong et al. [26] showed that soil exchangeable and soluble Ca were significantly greater in FRR + BC than in other treatments. In fact, the large Ca and Mg contents can cause the formation of Ca and Mg bound P [53,54]. Therefore,

a significant increase of Ca-Pi in FRR + BC can be explained by the input of Ca-Pi in biochar and the increase of soluble and exchangeable $Ca^{2+}$ in soil.

Our study showed that biochar, but not compost, incorporation into soils increased the total soil C, as has been reported previously [26,55]. However, soil labile C was not increased by either form of amendment. This phenomenon is consistent with the results of a previous study [56], in which there was no significant increase in labile C induced by a straw biochar added at rates of 10 to 40 Mg ha$^{-1}$. This is likely due to the small initial concentrations of labile C in the biochar and compost (Table 1). Moreover, the C mineralization rates in the soils amended with biochar or compost decreased dramatically after application to soil, and became similar to those in the control treatment, indicating rapid consumption of labile C contained in the biochar and compost in the early days [26,57]. Consequently, it was considered that labile C of FRR + Comp and FRR + BC was consumed soon after application and then only non-labile stable fractions remained in the soils. Indeed, this was confirmed by the significant increase in non-labile C of FRR + BC. These findings suggest that biochar amendment enhanced sequestration of non-labile C [17,58,59] which can contribute to soil quality in the long-term.

Contrary to our hypothesis, biochar amendment did not increase available N and total N in both sites, as was found of total N in the study by Phuong et al. [26]. In their study, N$_2$O emissions were significantly greater in FRR + Comp than in FRR and FRR + BC, indicating that the risk of N loss as N$_2$O was greater in compost than in biochar [3]. Besides, there was likely sufficient available N within the compost, but its concentrations were negligible in biochar [26]. Consequently, these results indicated that N in biochar and compost amendments might not play a major role in promoting grain yield, which was supported by Xie et al. [60] who reported that low bio-availability of biochar N did not make a significant impact on rice production or N nutrition of rice.

*4.2. Response of Enzyme Activities to Biochar or Compost Amendments to Fertilized Salt-Affected Acid Soils*

Greater ACP and ALP activities were observed in the FRR + BC of both sites at the tillering stage. Previous studies have shown that biochar amendment can alter the enzymatic activities in the soil, acting on the hydrolysis of organic phosphate compounds, with a consequent increase in the available P in soil and improved nutritional quality status of plant [61,62]. These results highlighted that increased ACP activity in the soil led to an increase in plant-available P, which indicates that ACP is a sensitive soil quality indicator of changes in soil management. This finding is also in agreement with Ahmed et al. [63] who reported that ACP was significantly related with labile organic P (NaHCO$_3$-Po), moderate labile organic P (NaOH-Po + HCl dissolved Po), or non-labile organic P (HCl insoluble Po).

Contrary to the tillering stage, most enzyme activities did not significantly increase with either compost or biochar amendment at the harvest. This result was similar to the previous study reported by Elzobair et al. [64] who found that the application of hardwood biochar (22.4 Mg ha$^{-1}$) to an Aridisol did not affect activities of several soil enzymes. As discussed above, labile C was not affected by compost or biochar amendment in this study. GLU is an extracellular enzyme mediating C mineralization [17]. According to Phuong et al. [3], biochar and compost amendments did not affect C mineralization after 3-day incubations. These suggested that FRR + Comp and FRR + BC did not have enough available C for C mineralization which requires GLU activity. In addition, watering was stopped 15 days before harvest and the soil moisture content decreased over time. In the year tested in this study, there was the lowest rainfall in December to January, and no rain in January at Kien Giang (Tables S1 and S2). Another study confirmed–unsurprisingly–that low soil moisture contents can strongly limit enzyme activity in soils [65]. Moreover, root growth would have ceased at the harvest stage, and thus there might be less competition with soil microorganisms. Besides, Hedley et al. [6] showed that soil phosphatase activity originates from plant roots. Therefore, we postulate that the reduced enzyme activities are explained by the low soil moisture content, curtailed root growth, and no differences

in labile C and available N. This is consistent with the previous studies, which showed that enzyme activities are driven primarily by labile C [66], reduced root activity [66], and moisture [67].

### 4.3. Relationship between Rice Yield and Biochemical Properties

Phuong et al. [26] showed at the fifth crop that biochar or compost application only improved straw biomass but not for grain yield at Ben Tre. In our study presenting the results of the sixth crop after 2 years of application of biochar and compost, the grain rice yield in both FRR + BC and FRR + Comp increased by 10% to 20% compared to RRR. Hence, the amendments were effective as nutrient fertilizers. This result is in agreement with previous studies that increase of rice yield following biochar application was related to increasing duration of application (over several cropping cycles), which may be ascribed to an improvement in soil fertility [68–71]. Our studies showed that the observed positive effect on rice yield was related to generation of labile P followed by moderate labile P, ACP, and total C. Thus, although the rates of biochar amendment over six crops (50 Mg ha$^{-1}$) are determined to improve the salinity level in soil and subsequent rice growth, the incorporation of biochar simultaneously increased the amount of P available to crops and increased the grain yield. Rafiq et al. [72] reported that combined application of chemical fertilizers with manure changed the P fractions and increased grain yield by improving soil chemical properties. Similarly, Wu et al. [73] showed that straw biochar had a potentially more positive impact on rice productivity and P retention than straw returning in saline–alkaline soil.

Al-P, Fe-P, and Ca-P are among the moderately labile P pool [63]. Sushanta et al. [74] showed that available P and Al-P were positively and significantly correlated with grain yield with an increasing level of P from 60 kg P$_2$O$_5$ ha$^{-1}$ to 120 kg P$_2$O$_5$ ha$^{-1}$. Although these were immediately available for plants, these had the potential to become available after a relatively slow process because there is a dynamic balance between available P and non-available P. For instance, a part of NaOH-Pi, which is contained in FRR + BC and FRR + Comp treatments at a high content, might be transformed to available P by means of solubilizing Fe and Al phosphates by increased pH [3]. Another possible mechanism is that P reduces the concentrations of free Al$^{3+}$ and Fe$^{3+}$ by precipitating with P, which might be related to better rice root growth resulting in more nutrient absorption.

### 4.4. Triple versus Double Rice Cropping

Where organic materials were not applied, all measured parameters–including rice yield–were apparently insensitive to a triple or double rice crop in one year. The exception of a single instance of the activity of one enzyme is unlikely to be consequential. Hence, the inclusion of a fallow period in the cropping cycle did not have a major effect on the system, at least in terms of the scope of our study. This suggests that at least over a two-year period in the inorganically fertilized subsystem, the more crop-intensive management regime in these soils did not impair soil quality. However, yield was clearly still constrained by a lack of P, since P fertilization arising from biochar resulted in more grain.

### 5. Conclusions

This study provides further evidence that biochar addition to fertilized salt-affected acid soils can contribute to P fertilization of paddy rice crops and increase grain rice yield. This is likely via affecting the partitioning of P forms between those more available to plants, with some evidence for a contribution by enzyme action. A further study is now ongoing to monitor P leaching risk under long-term biochar and compost application to paddy rice soil.

**Supplementary Materials:** The following supporting information can be downloaded at: https://www.mdpi.com/article/10.3390/agronomy13061593/s1, Table S1. Rainfall and temperature at Thanh Phu district, Ben Tre province. Table S2. Rainfall and temperature at U Minh Thuong district, Kien Giang province.

**Author Contributions:** D.T.T.L. conceptualized the study; T.B.L., D.D.M., H.M.T.M. and T.T.L. conducted the experiments and collected samples; D.T.T.L., N.V.S., N.T.K.P. and H.M.T.M. performed sample analysis and data analysis; D.T.T.L. and K.T. validated data; D.T.T.L. wrote the paper—original draft preparation; D.T.T.L., C.M.K., K.R., N.V.S., N.T.K.P. and K.T. reviewed, edited, and finalized the paper. All authors have read and agreed to the published version of the manuscript.

**Funding:** The Can Tho University Improvement Project VN14-P6 provided partial funding for this study, with support from a loan granted by the Japanese ODA.

**Data Availability Statement:** Not applicable.

**Acknowledgments:** We are truly grateful to the A-8 project team members at the College of Soil Science, Can Tho University, for their support during the field experiments.

**Conflicts of Interest:** The authors declare no conflict of interest.

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
