# Peer review of "Effects of Rice Husk Biochar and Compost Amendments on Soil Phosphorus Fractions, Enzyme Activities and Rice Yields in Salt-Affected Acid Soils in the Mekong Delta, Viet Nam"

_agronomy, doi:10.3390/agronomy13061593_

Round 1
Reviewer 1 Report
Compost and biochar have been proved to present high potentials in enhancing the ability of plants endure adverse environmental stresses and increasing crop production. The addition of organic amendments to soils can supply essential nutrients for plant growth; promote a reduction in electrical conductivity, the exchange of Na+ with Ca2+, and the formation of large water-stable aggregates; improve soil permeability and water movement; and provide habitats and organic carbon for soil microorganisms. The present study studied the beneficial effects of biochar and compost on various soil phosphorus fractions, enzyme activities and yields of rice. Overall, the MS presents an excellent work. The methodology is adequate, the results are clearly presented, and the discussion is appropriate. I don’t have questions except the Conclusions section is too concise to provide valuable information.
Author Response
Compost and biochar have been proved to present high potentials in enhancing the ability of plants endure adverse environmental stresses and increasing crop production. The addition of organic amendments to soils can supply essential nutrients for plant growth; promote a reduction in electrical conductivity, the exchange of Na+ with Ca2+, and the formation of large water-stable aggregates; improve soil permeability and water movement; and provide habitats and organic carbon for soil microorganisms. The present study studied the beneficial effects of biochar and compost on various soil phosphorus fractions, enzyme activities and yields of rice. Overall, the MS presents an excellent work. The methodology is adequate, the results are clearly presented, and the discussion is appropriate. I don’t have questions except the Conclusions section is too concise to provide valuable information.
Reply: Thank you very much for your positive suggestions to our manuscript. The style of Conclusion is diverse. In Conclusion, we would like to focus only the essence of this manuscript. We consider this is also a style of Conclusion. However, based on the comment, we added a sentence describing a future direction.
Reviewer 2 Report
This study is very meaningful and has achieved sounded results. I suggest merging the two bar charts together and labeling the treatment, site, and their interaction effects in Figure 3.
Author Response
This study is very meaningful and has achieved sounded results. I suggest merging the two bar charts together and labeling the treatment, site, and their interaction effects in Figure 3.
Reply: Thank you very much for the suggestion. We combined two figures in Figure 3 in the revised manuscript.
Reviewer 3 Report
The weakest part of the article is the methodology - several parameters of the tested materials are missing.
The authors did not realize that they were evaluating the interaction effect of biochar and mineral NPK fertilizers and not the effect of biochar alone.
After acceptance comments, recommendations, the article can be considered for publication.

Author Response
The weakest part of the article is the methodology - several parameters of the tested materials are missing.
The authors did not realize that they were evaluating the interaction effect of biochar and mineral NPK fertilizers and not the effect of biochar alone.
After acceptance comments, recommendations, the article can be considered for publication.
Reply: Thank you very much for your constructive and useful comments. We described our responses below.
Line 40: Please distinguish the labels - total phosphorus, inorganic phosphorus, organic phosphorus, accessible phosphorus.
Not all inorganic P in the soil is in an accessible form for plants. I recommend leaving the letter P for the designation of accessible phosphorus, as is customary, and using the designation Pt (total phosphorus) and the like for total phosphorus.
Reply: Thank you for the useful comment. We also consider that description of P is based on the way the reviewer suggested. Only exception is that we describe available P as accessible phosphorus. For further clarification, we changed from “P forms in (line 44)” to “total P forms”.
Line 90: Explain the acronym MRD (abbreviation MRD)
Reply: Thank you very much for the comment. This was our mistake and we did not show explanation for MR and MRD. We added the full spelling for MD (Mekong Delta) and unified with MD.
Line 95: Explain the acronym MD
Reply: please see the above comment (Line 90)
Line 105: Add to chapter 2.3 whose method, or what methodical procedure did you use to determine soluble Ca and how did you determine exchangeable Ca.
Reply: Thank you very much for the comment. Since all the methods are already described in our previous manuscript, we reduced the related descriptions. However, in the revised manuscript, we added some more explanations.
Line 106: Enter the name of the company that produces the biochar you used.
Reply: Thank you very much for the comment. Since the information is already described in our previous manuscript, we omit it. However, in the revised manuscript, we added the name of the company.
Line 112: The second part of the sentence is unnecessary. Do not mention it.
Reply: Thank you very much for the comment. We deleted the relevant part.
Table 1: Give the following required data: Contents of inorganic nitrogen, content of accessible (available) P and K. C:N ratios, pH values, possibly other parameters and at the same time give names of the methodology for determining these parameters.
The principle applies. When I test the influence of a material, I must have the parameters of the tested material defined. Then I can anticipate its influence.
At the same time, the following applies: if I am testing the effect of a material on parameter X in the soil or plant, then I must determine the same parameter X in the tested material.
Add to table 1 the values of those parameters that you present in tables 3 and 4.
Reply: Thank you very much for the comment. Since many basic parameters are already described in our previous manuscript, we described the results that were not reported in our previous manuscript. However, in the revised manuscript, we added related results and mentioned that some of the results are cited from our previous manuscript.
Line 155: Were mineral NPK fertilizers applied to all variants? If so, then you should talk about the interaction effect of compost and NPK fertilizers, or on the interaction effect of biochar and NPK fertilizers.
Reply: see the below comment. (Line 351)
Line 194: When determining total N, in addition to the instrument used, also state the author of the methodology, or at least state the basic principle of the determination.
Reply: Thank you very much for the comment. We added the relevant information in the revised manuscript.
Line 214: A large number of abbreviations does not contribute to the clarity of the text.
Reply: Thank you very much for the valuable comment and we agree with the reviewer. In the revised manuscript, we stopped using abbreviation names for the research sites.
Table 3: You did not give a reason why in the BT soil the application of compost tended to acidify the soil and in the KG soil it tended to alkalize it.
Reply: Thank you very much for the comment. Our results are exactly the same as the reviewer pointed out. However, we do not reach to find a definitive reason and remained not to describe the reason.
Line 351: I repeat and emphasize. This is an interaction effect and not an effect of Biochar and compost.
If you did not use NPK fertilizers, the effect of biochar would be different.
Usually, the application of pure biochar, without mineral or organic fertilizers, reduces yields (we also have our own results). For this reason, it is necessary to restyle the title of the article, the titles of subsections and also the conclusions.
Reply: Thank you very much for the valuable comment. We agree with the comment that the application of pure biochar without mineral or organic fertilizers reduces yield. Thus, considering practical application, we never think that biochar will be applied to a paddy field without fertilizers. In this study, we added biochar or compost as a supplement to the conventional chemical fertilizer to increase soil quality and thereby rice yield. However, we also agree with the comment that “this is an interaction effect”. We changed the titles of subsections and conclusion from “biochar and compost amendments” to “biochar and compost amendments to fertilized salt-affected acid soils” and from “salt-affected acid soils” to “fertilized salt-affected acid soils”, respectively.
Reviewer 4 Report
The manuscript entitled ‘’Effects of Rice Husk Biochar and Compost Amendments on Soil Phosphorus Fractions, Enzyme Activities and Rice Yields in Salt-Affected Acid Soils in the Mekong Delta, Viet Nam’’ is an interesting work with valuable findings. However, there are some important points need to be improved before further process.
Line 61: ‘’Thus, it is important to explore how biochar or compost addition affects…’’ -> Until here, there are nothing about compost application in rice field. Also there are some published paper as well that compared biochar and compost in one study. Authors need to bring some information about background of the idea of biochar and compost comparison. Here is one you can add it here: https://doi.org/10.1007/s10333-022-00912-8. Please add some regard to comparison between these two materials.
Section 2.1. Materials description/Table 1 -> Please bring some more information about intrinsic characteristics of both, specially biochar. Parameters like CHNO, Ash content, surface area, CEC, … > if available!
-Please add information about your pyrolysis condition as well! How the biochar produced? Heating rate? Which kind of device?
Conclusions: please extend this section with some futuristic suggestion for further work in this topic.
Good luck!
Author Response
The manuscript entitled ‘’Effects of Rice Husk Biochar and Compost Amendments on Soil Phosphorus Fractions, Enzyme Activities and Rice Yields in Salt-Affected Acid Soils in the Mekong Delta, Viet Nam’’ is an interesting work with valuable findings. However, there are some important points need to be improved before further process.
Line 61: ‘’Thus, it is important to explore how biochar or compost addition affects…’’ -> Until here, there are nothing about compost application in rice field. Also there are some published paper as well that compared biochar and compost in one study. Authors need to bring some information about background of the idea of biochar and compost comparison. Here is one you can add it here: https://doi.org/10.1007/s10333-022-00912-8. Please add some regard to comparison between these two materials.
Reply: thank you very much for the valuable comment. Based on the comment, we added one sentence in line 60 to 61 and cited the recommended reference. “Among organic amendments, biochar and compost are two common materials.”
Section 2.1. Materials description/Table 1 -> Please bring some more information about intrinsic characteristics of both, specially biochar. Parameters like CHNO, Ash content, surface area, CEC, … > if available!
Reply: Thank you very much for the comment. Since many basic parameters such as total C and N, surface area and CEC, are already described in our previous manuscript, we described the results that were not reported in our previous manuscript. However, in the revised manuscript, we added related results and mentioned that some of the results are cited from our previous manuscript.
-Please add information about your pyrolysis condition as well! How the biochar produced? Heating rate? Which kind of device?
Reply: Thank you very much for the comment. Some pyrolysis conditions are already reported in our previous manuscript and therefore we did not mention. In the revised manuscript, we added information on the pyrolysis temperature. However, we do not have information on the heating rate and device used for its production.
Conclusions: please extend this section with some futuristic suggestion for further work in this topic.
Reply: Thank you very much for your comment. We added a sentence describing future direction.
Round 2
Reviewer 3 Report
I have included comments and observations on the article both in the margin of the article and directly in the review. Therefore, I expected that the authors would respond to each comment individually in writing. The authors incorporated the corrections directly into the text. They did not respond in writing to every comment made in the review. So it happened that some comments were accepted by the authors and some were not.
Author Response
Second review by Reviewer 3.
Comments: I have included comments and observations on the article both in the margin of the article and directly in the review. Therefore, I expected that the authors would respond to each comment individually in writing. The authors incorporated the corrections directly into the text. They did not respond in writing to every comment made in the review. So it happened that some comments were accepted by the authors and some were not.
Reply: Thank you very much for your valuable comments. Yes, we noticed that Reviewer 3 pointed out both in the reviewer sheet and in the margin of the article. We picked up all the comments directly written in the margin and replied in the previous answer sheet. Therefore, we think that Reviewer 3 misunderstand our previous reply. We just repeat the previous answer sheet below.
Comments by Reviewer 3
The weakest part of the article is the methodology - several parameters of the tested materials are missing.
The authors did not realize that they were evaluating the interaction effect of biochar and mineral NPK fertilizers and not the effect of biochar alone.
After acceptance comments, recommendations, the article can be considered for publication.
Reply: Thank you very much for your constructive and useful comments. We described our responses below.
Line 40: Please distinguish the labels - total phosphorus, inorganic phosphorus, organic phosphorus, accessible phosphorus.
Not all inorganic P in the soil is in an accessible form for plants. I recommend leaving the letter P for the designation of accessible phosphorus, as is customary, and using the designation Pt (total phosphorus) and the like for total phosphorus.
Reply: Thank you for the useful comment. We also consider that description of P is based on the way the reviewer suggested. Only exception is that we describe available P as accessible phosphorus. For further clarification, we changed from “P forms in (line 44)” to “total P forms”.
Line 90: Explain the acronym MRD (abbreviation MRD)
Reply: Thank you very much for the comment. This was our mistake and we did not show explanation for MR and MRD. We added the full spelling for MD (Mekong Delta) and unified with MD.
Line 95: Explain the acronym MD
Reply: please see the above comment.
Line 105: Add to chapter 2.3 whose method, or what methodical procedure did you use to determine soluble Ca and how did you determine exchangeable Ca.
Reply: Thank you very much for the comment. Since all the methods are already described in our previous manuscript, we reduced the related descriptions. However, in the revised manuscript, we added some more explanations.
Line 106: Enter the name of the company that produces the biochar you used.
Reply: Thank you very much for the comment. Since the information is already described in our previous manuscript, we omit it. However, in the revised manuscript, we added the name of the company.
Line 112: The second part of the sentence is unnecessary. Do not mention it.
Reply: Thank you very much for the comment. We deleted the relevant part.
Table 1: Give the following required data: Contents of inorganic nitrogen, content of accessible (available) P and K. C:N ratios, pH values, possibly other parameters and at the same time give names of the methodology for determining these parameters.
The principle applies. When I test the influence of a material, I must have the parameters of the tested material defined. Then I can anticipate its influence.
At the same time, the following applies: if I am testing the effect of a material on parameter X in the soil or plant, then I must determine the same parameter X in the tested material.
Add to table 1 the values of those parameters that you present in tables 3 and 4.
Reply: Thank you very much for the comment. Since many basic parameters are already described in our previous manuscript, we described the results that were not reported in our previous manuscript. However, in the revised manuscript, we added related results and mentioned that some of the results are cited from our previous manuscript.
Line 155: Were mineral NPK fertilizers applied to all variants? If so, then you should talk about the interaction effect of compost and NPK fertilizers, or on the interaction effect of biochar and NPK fertilizers.
Reply: see the below comment.
Line 194: When determining total N, in addition to the instrument used, also state the author of the methodology, or at least state the basic principle of the determination.
Reply: Thank you very much for the comment. We added the relevant information in the revised manuscript.
Line 214: A large number of abbreviations does not contribute to the clarity of the text.
Reply: Thank you very much for the valuable comment and we agree with the reviewer. In the revised manuscript, we stopped using abbreviation names for the research sites.
Table 3: You did not give a reason why in the BT soil the application of compost tended to acidify the soil and in the KG soil it tended to alkalize it.
Reply: Thank you very much for the comment. Our results are exactly the same as the reviewer pointed out. However, we do not reach to find a definitive reason and remained not to describe the reason.
Line 351: I repeat and emphasize. This is an interaction effect and not an effect of Biochar and compost.
If you did not use NPK fertilizers, the effect of biochar would be different.
Usually, the application of pure biochar, without mineral or organic fertilizers, reduces yields (we also have our own results). For this reason, it is necessary to restyle the title of the article, the titles of subsections and also the conclusions.
Reply: Thank you very much for the valuable comment. We agree with the comment that the application of pure biochar without mineral or organic fertilizers reduces yield. Thus, considering practical application, we never think that biochar will be applied to a paddy field without fertilizers. In this study, we added biochar or compost as a supplement to the conventional chemical fertilizer to increase soil quality and thereby rice yield. However, we also agree with the comment that “this is an interaction effect”. We changed the titles of subsections and conclusion from “biochar and compost amendments” to “biochar and compost amendments to fertilized salt-affected acid soils” and from “salt-affected acid soils” to “fertilized salt-affected acid soils”, respectively.